# CodeAlign: Resolving Modality Isolation in Heterogeneous Collaborative Perception

## Abstract

Collaborative perception leverages data exchange among multiple agents to overcome the perception limitation of individual agents, significantly enhancing overall perception capabilities. However, heterogeneity brings domain gaps among agents, hindering the collaboration. The heterogeneity is further compounded by an underexplored problem, modality isolation, where the absence of co-occurring data across certain modalities leads to even bigger domain gaps and limits feature alignment approaches. To address this problem, we propose CodeAlign, the first framework to systematically resolve modality isolation in heterogeneous collaborative perception. The key idea is to partition modalities into groups based on whether they have isolation or not, and apply customized strategies for intra-group and inter-group alignment. For intra-group alignment, CodeAlign introduces *code space formation* that constructs a shared discrete feature space using a codebook, enabling effective feature alignment and efficient communication. For inter-group alignment, CodeAlign introduces *code space translation* that establish mappings between code spaces, facilitating efficient and dynamic feature transfer. A lightweight Unified Code Translator is designed to perform convenient one-to-many code translation, controlled by conditional embeddings. Experiments show that CodeAlign reduces training parameters by 92% when integrating 4 new modalities, and achieving 1024× lower communication volume, while maintaining on-par perception performance with SOTA methods. The code will be released.

## 1 Introduction

Collaborative perception plays a pivotal role in intelligent systems such as connected autonomous vehicles and multi-robot collaboration. It enables agents to build a more comprehensive understanding of the environment by sharing perceptual information. In real-world applications, however, vehicles from different manufacturers often exhibit heterogeneity leading to significant domain gaps during feature-level collaboration. Heterogeneity includes different sensor types, sensor parameters, and perception models. Late fusion bypasses heterogeneity by integrating detection outputs, but suffers from suboptimal performance, localization noise Lu et al. (2022), and communication latency Wang et al. (2020). Traditional approaches often use fusion networks for collective training on data involving collaborating modalities Xiang et al. (2023). Further developments leverage contrastive learning between intermediate features to facilitate feature alignment Luo et al. (2024). To enable extensible heterogeneous collaboration, some approaches generate standardized intermediate features, using contrastive learning to align modalities toward this common representation for easier integration of new modalities Gao et al. (2025). These methods all require collaborative data for training, that is, data from different modalities must have cooperated within the same scene.

A critical yet understudied issue in heterogeneous settings is **modality isolation**, illustrated in Figure 1. Perception data are collected by different institutions across diverse locations and times, resulting in datasets that each cover only limited modalities. Consequently, many modalities lack co-occurring data from the same scene, meaning they have never collaborated in any recorded data, and are more difficult in alignment. We refer to this situation as modality isolation. Modality isolation significantly increasing the difficulty of achieving robust and generalizable alignment. When aligning two modality-isolated agents, the absence of co-occurring data in any frame makes it impossible to establish mutual supervision through correspondence of BEV features Gao et al. (2025), nor can shared ground truth data be utilized as reference to facilitate alignment. For collective

trained fusion networks, it is feasible to alternately input single-modality data from isolated modalities; however, this significantly impairs perception performance. Although (Lu et al., 2024) adapts to modality-isolated scenarios through a extension strategy trained on local data of each modalities, it is limited by the high training cost and inconvenience of retraining encoders.

Modality isolation post the following challenges: i) Heterogeneity arises from diverse factors, resulting in a wide spectrum of heterogeneous types. ii) Continuous technological advancements lead to the emergence of new modalities, which inevitably suffer from modality isolation with existing ones. iii) Datasets collected by different institutions are often subject to data privacy requirements, further restricting data accessibility. Under such conditions, the system must be capable of aligning numerous modalities efficiently, while maintaining extensibility and protecting data privacy.

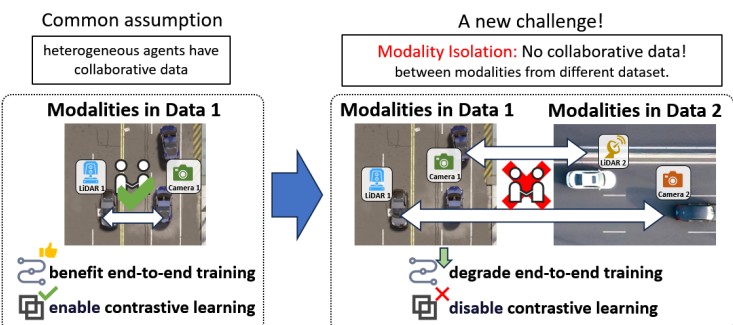

Figure 1: Illusion of modality isolation in heterogeneouse collaborative perception.

To address this challenge, we propose CodeAlign, the first framework to systematically resolve modality isolation in heterogeneous collaborative perception. The key idea is a group-wise alignment strategy, which groups non-isolated modalities and separates isolated ones, applying specialized strategies for both intra- and inter-group alignment. Group-wise alignment is crucial for training efficiency. Without grouping, each modality must align with every other, since collective training is ineffective under modality isolation, leading to high training costs. Grouping reduces complexity while enabling non-isolated modalities to fully leverage collaborative data for stronger perception.

To achieve efficient group-wise alignment, CodeAlign involves two key steps: i) Intra-group alignment is accomplished through Code Space Formation. The step inserts a shared learnable codebook into the perception framework, establishing a discrete shared feature space within the group. This maintains the invariance of the network while reducing training parameters. ii) Inter-group alignment is achieved by Code Space Translation through a proposed lightweight Unified Code Translator. The model learns the mapping from the current group's code space to multiple groups' feature code spaces, incorporating different conditional embeddings for different mappings by gated modulation. The training of the code translation model uses only the local data of the current group, ensuring data privacy. By leveraging the discrete nature of codebook, CodeAlign not only reduces the complexity of feature space formation and mutual mapping between feature spaces, but also compresses information, significantly improving communication efficiency. Compared with existing methods, CodeAlign overcomes the challenge of modality isolation by eliminating the reliance on co-occurring data, while achieving reduced training costs and communication overhead.

We evaluate CodeAlign on the OPV2V Xu et al. (2022d) dataset and demonstrate significant improvements in efficiency. Our method reduces training parameters by 92% when four new agent types are added, while maintain well-matched perceptual performance compared to SOTA approaches. Furthermore, CodeAlign reduces communication volume by 1024×, highlighting its practical utility in bandwidth-constrained environments. Our contributions are as follows:

- We propose CodeAlign, the first framework that systematically tackles modality isolation in heterogeneous collaborative perception, which improves extensibility and efficiency.

- CodeAlign introduce a group-wise alignment paradigm for modality isolation. We present a Shared Code Space Formation step for effective intra-group alignment and a Unified Code Translator for efficient inter-group alignment.

- Extensive experiments demonstrate that CodeAlign reduces training parameters by 99% when integrating three new agent types and achieves 1024× lower communication volume, outperforming existing methods in accuracy.

## 2 RELATED WORKS

### 2.1 COLLABORATIVE PERCEPTION

Collaborative perception improves detection accuracy by leveraging shared sensory information across multiple agents and is commonly classified into early, intermediate, and late fusion strategies. Early fusion transmits raw sensor data, incurring high communication cost, while late fusion shares only bounding boxes, limiting performance and robustness due to feature loss. Intermediate fusion (Hu et al., 2022; Fu et al., 2025) has gained popularity for achieving a favorable balance between performance and communication efficiency. To advance research in multi-agent collaborative perception, OPV2V (Xu et al., 2022d) provides simulated vehicle-to-vehicle collaboration, DAIR-V2X (Yu et al., 2022) offers real-world vehicle-infrastructure data, and RCooper (Hao et al., 2024) introduces adverse weather conditions for robustness evaluation. To mitigate communication bottlenecks, Hu et al. proposed Where2comm (Hu et al., 2022) and CodeFilling (Hu et al., 2024), which reduce redundancy in transmitted features. Other approaches (Wei et al., 2023; Lei et al., 2022) address communication disruptions or latency by exploiting historical interaction data or temporal context. Despite these advances, most existing methods assume homogeneous sensor modalities and identical models across agents. In this work, we study heterogeneous collaborative perception, where agents may possess different sensor and model configurations. We further identify and investigate the underexplored challenge of modality isolation, arising when heterogeneous agents struggle to effectively align and fuse features due to significant domain gaps.

### 2.2 HETEROGENEOUS COLLABORATIVE PERCEPTION

Heterogeneity in collaborative perception arises from differences in sensor modalities, sensor configurations, and perception model architectures. Early works focus on LiDAR-based heterogeneity: V2XViT (Xu et al., 2022c) addresses spatial misalignment between vehicle and infrastructure; MPDA (Xu et al., 2022b) and Calibrator (Xu et al., 2022a) study heterogeneous LiDAR models; PNPDA (Luo et al., 2024) further considers varying voxel sizes; and PolyInter (Xia et al., 2025) explores scalability. However, the domain gap between LiDAR and camera data constitutes a more significant challenge. HMViT (Xiang et al., 2023) proposes collective training of cross-modal models to bridge this gap but lacks scalability. STAMP (Gao et al., 2025) train a network to provide reference features for the scene, and leverage contrastive learning to align heterogeneous features to these references, achieving scalable heterogeneous cooperative perception. However, a new challenge—modality isolation—arises from the fact that different modalities are rarely co-collected in the same scenes, leading to a lack of co-occurring training data. This limitation hinders effective end-to-end training and renders contrastive learning infeasible. HEAL addresses this issue through backward alignment by retraining encoders on local data, but suffers from high computational and communication overhead. In this work, we propose CodeAlign, a group-wise alignment framework under modality isolation that leverages code space construction and mapping to systematically address this challenge.

## 3 METHODOLOGY

### 3.1 PRELIMINARY: GROUP-WISE ALIGNMENT PARADIGM

Group-wise alignment consists of two steps: intra-group alignment and inter-group alignment. Intra-group alignment jointly trains modalities that have collaborative data to establish a shared, aligned feature space, thereby fully enhancing collaboration performance and reducing the number of required alignments. Inter-group alignment connects the pre-trained groups by establishing mappings to enable inter-group collaboration. Let the observation of agent $k$ be $O_k$, and define ego group and other groups as collection $G_e$ and $G_n$. Then, the detection output $B_i$ from ego agent $i$ is obtained as:

$$F_k = f_{\text{enc}[k]}(O_k), \quad k \in G_e \cup G_n \tag{1}$$

$$\begin{cases} M_k = f_{\text{align}[G_e]}(F_k), & k \in G_e \\ M_k = f_{\text{trans}[G_n G_e]}(F_k), & k \in G_n \end{cases} \tag{2}$$

$$F_{k \to i} = \Gamma_{k \to i}(M_k) \tag{3}$$

$$\tag{4}$$

$$H_i = f_{\text{fusion}}\left(\{F_{k \to i}\}_{k \in G_e \cup G_n}\right) \tag{5}$$

$$B_i = f_{\text{head}}(H_i) \tag{6}$$

where $F_k$ is the encoded feature of each agent, and $M_k$ is the aligned feature obtained either from alignment module $f_{align[\cdot]}(\cdot)$ of ego group or transformation module $f_{trans[\cdot]}(\cdot)$ of other groups. $\Gamma_{k \to i}$ is the spatial transformation applied during message passing ($F_{i \to i} = F_i$). Fusion network $f_{fusion}(\cdot)$ integrates all transformed features to the fused feature $H_i$, and detection head $f_{head}(\cdot)$ output the detection results.

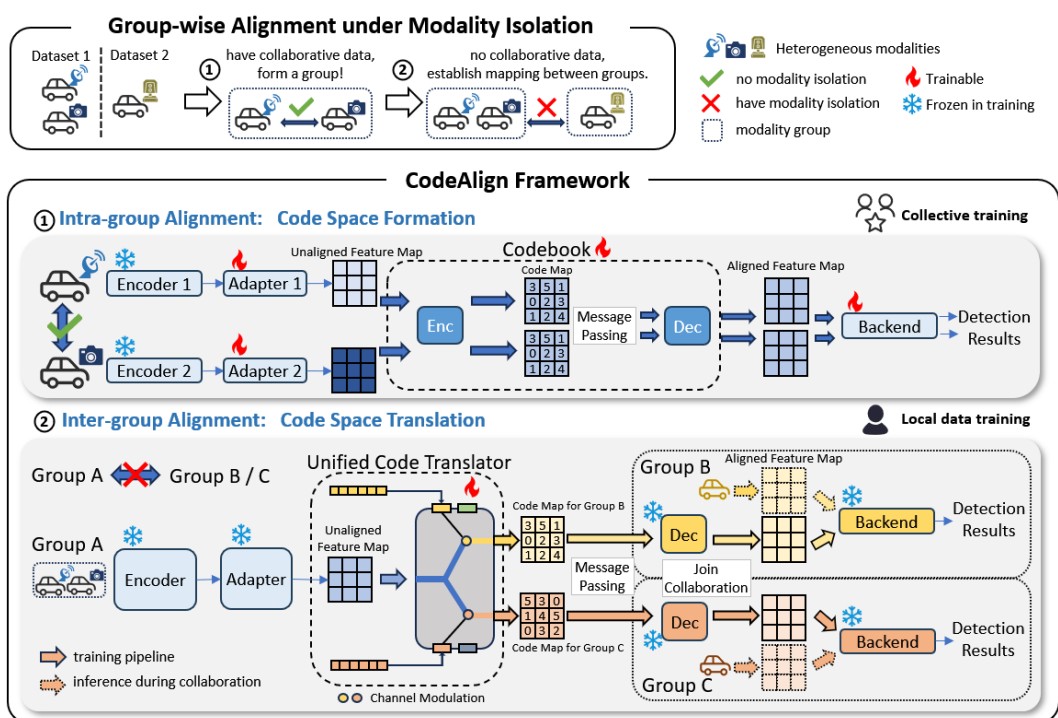

Figure 2: Overview of CodeAlign framework. The form of group-wise alignment is described above the figure. CodeAlign takes discrete code space as the core and includes intra-group alignment and inter-group alignment. Alignment is achieved through the formation and translation of code space.

## 3.2 CODEALIGN FRAMEWORK

To achieve efficient group-wise alignment under modality isolation, CodeAlign consists of two key steps, as illustrated in Figure 2: i) For intra-group alignment, Shared Code Space Formation utilizes a codebook to construct a discrete shared feature space for each group. ii) For inter-group alignment, a Unified Code Translator maps features into codes of multiple target code spaces, with the transformation controlled by conditional embeddings through channel modulation. Compared to the common group-wise alignment, the equation 2 and equation 3 in CodeAlign framework can be expressed as:

$$\begin{cases} I_k = \Phi(C_{[G_e]Enc}(F_k)), & k \in G_e \\ I_k = T_{[G_n]}(F_k, E_{[G_n]to[G_e]}), & k \in G_n \end{cases} \tag{7}$$

$$F_{k \to i} = \Gamma_{k \to i}(C_{[G_e]Dec}(I_k)) \tag{8}$$

where $C_{[G_e]}$ is the codebook of group $G_e$, while $C_{[G_e]Enc}$ and $C_{[G_e]Dec}$ are encoder and decoder of codebook. $I$ is the code map transformed by calculator $\Phi$. $T_{[G_n]}$ is the Unified Code Translator of group $G_n$, and $E_{[G_n]to[G_e]}$ is the conditional embedding target at group $G_e$, used to control the translator's output.

### 3.2.1 INTRA-GROUP ALIGNMENT: SHARED CODE SPACE FORMATION

The purpose of intra-group alignment is to align modalities that have collaborative data into a shared feature space, and prepare the feature space as representation of current group for later inter-group

alignment. In CodeAlign, a codebook is employed as the shared feature space (code space) of a group. The encoded BEV features are aligned into the code space by assigning each feature to the nearest embedding, yielding a compact code map composed of codebook indices for message passing. During communication, only the code map is transmitted and subsequently decoded into the corresponding embeddings. Since the encoded BEV features are mapped from the same shared code space, effective alignment is achieved. For instance, features corresponding to a vehicle from different modalities may be represented by the same embedding in the code space, resulting in identical decoded features, which facilitates seamless fusion.

In perception systems, the encoder extracts features from raw inputs, and is deeply coupled with downstream tasks. Arbitrary modifications can compromise the overall system performance and stability. HEAL achieves alignment by retraining the encoder, which disrupts the consistency of the perception system and introduces additional training overhead. In contrast, CodeAlign preserves the encoder in a frozen state during alignment, inserting the adapter and codebook only at intermediate stage of the framework, thereby maintaining the integrity and modularity of the overall system.

**Multi modality grouping.** As shown in Fig. 2, for modalities within a group that have access to collaborative data, a learnable codebook is inserted into the perception pipeline between the encoder and the backend to establish an effective and efficient shared code space. The backend is shared and trainable, as a network processing features from the code space should be learned. A lightweight ResNet-based adapter is introduced before the codebook to accelerate the alignment. Based on the shared codebook $C$, each agent can replace the encoded feature map $F_k$ with a series of code indices $I_k$, forming a compact code map. For each BEV location $(h, w)$, the code index is computed by calculator $\Phi$ as,

$$\Phi: \quad (I_k)_{[h,w]} = \arg\min_\ell \left\| (C_{[G_e]Enc}(F_k))_{[h,w]} - C[\ell] \right\|_2^2 \tag{9}$$

The code map is used for message passing and is decoded by $C_{[G_e]Dec}$ to reconstruct the aligned feature map, as the aligned features maps are composed of deterministic discrete features in the code space. The transmitted intermediate feature is compressed from $H * W * C$ to $H * W * log2(D)$, where $D$ denotes the codebook size, significantly reducing communication bandwidth.

**Single modality grouping.** Homogeneous datasets are also widely available, and these single modalities can form standalone groups, enabling the efficient construction of a code space using a simplified approach. For homogeneous perception pipelines, both the encoder and the backend can be kept frozen, with only the adapter and codebook inserted and trained in the middle. This approach minimizes the number of trainable parameters, preserves the integrity of the original modules, and maintains most of the original perception performance.

**Loss Function.** To accelerate alignment and ensure consistent feature representations across agents, we supervise three objectives: object detection, fusion learning, and inter-agent feature similarity. The overall loss is defined as:

$$L = L_{\text{det}}\left(\widehat{\mathcal{O}}_i, \mathcal{O}_i^0\right) + L_{\text{pyramid}} + \lambda \sum_{M_w \neq M_j} L_{\text{sim}}(F_{w \to i}, F_{j \to i}), \quad i, w, j \in G_e \tag{10}$$

where $L_{\text{det}}(\cdot)$ denotes the detection loss, $\mathcal{O}_i^0$ and $\widehat{\mathcal{O}}_i$ represent the ground-truth and predicted object states for agent $i$, respectively. $L_{\text{pyramid}}$ is the pyramid loss from HEAL(), as we employ pyramid fusion as fusion net. $L_{\text{sim}}(\cdot)$ is similarity loss, which enforces feature consistency among collaborating agents. $M$ denotes modality, only similarity between different modalities is calculated. We adopt the cosine similarity between pairwise aligned features, defined as $L_{\text{sim}}(F_{w \to i}, F_{j \to i}) = (1 - \cos(F_{w \to i}, F_{j \to i}))$. During the optimization, the network parameters and the codebook are updated simultaneously.

### 3.2.2 INTER-GROUP ALIGNMENT: CODE SPACE TRANSLATION

Since each group has established its own shared code space to represent group-specific features, inter-group alignment requires a mechanism to translate between these heterogeneous code spaces. Within the CodeAlign framework, translation can occur between dense features or code maps. Among the feasible strategies, dense-to-dense (D2D) translation offers lossless transformation but incurs high computational cost and forfeits the bandwidth efficiency advantage. Conversely, code-to-code (C2C) translation suffers from excessive quantization error due to its highly discrete nature, causing significant information loss and degraded alignment performance. The dense-to-code

(D2C) approach achieves the optimal balance: it preserves the low-bandwidth communication benefits while maintaining reconstruction fidelity within acceptable limits, making it the most practical solution for inter-group alignment.

CodeAlign achieve inter-group alignment through Code Space Translation with a code translator in D2C paradigm. Given an encoded dense feature from a source group, the translator maps it into a code map defined by the target group's codebook. This compact representation is then transmitted and decoded using the target group's codebook decoder, allowing the feature to be reconstructed in the target group's code space and seamlessly integrated into its collaborative perception pipeline. The simplest implementation of a code translator is a one-to-one translator, where each pair of groups trains a translator for each other.

**Local Group Data Training.** The modality isolation issue highlights data privacy preservation, as datasets from different institutions may not be shared externally. To address this, we design a privacy-aware training protocol that relies exclusively on local data: the source group processes its own data through its encoder and the code translator, with the intermediate feature fed directly into the target

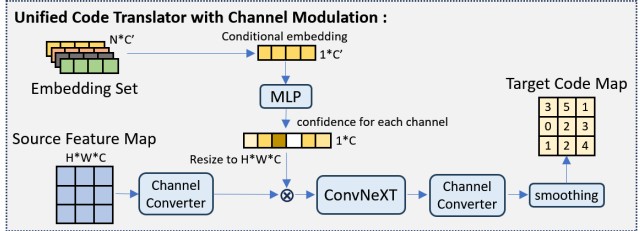

Figure 3: Model structure of the Unified Code Translator.

group's decoder and backend. The detection loss computed on the target backend's output serves as the supervision, explicitly encouraging the translator to produce features that align with the target group's code space. This approach requires no external data transmission, achieves effective alignment by local training with the ego group's data, and fully complies with data privacy regulations while enabling cross-institutional collaboration.

**Multi-group Translation.** In scenarios involving collaboration among multiple groups, the conventional one-to-one translation paradigm suffers from significant drawbacks: a complex training process and high inference memory usage. The training complexity arises because a dedicated translator must be trained for every possible pair of groups, and often in both directions. During inference, the system must load all trained translators into memory to be prepared for potential collaboration with any modality, leading to substantial memory overhead. A straightforward alternative is to train a shared backbone with multiple output heads, each dedicated to a specific target group. While this approach improves model reuse and reduces training cost, it still demands loading all output heads at inference time, making the memory footprint grow linearly with the number of groups.

To address this issue, we propose a lightweight Unified Code Translator, which operates through channel modulation with conditional embeddings. The training and inference process is illustrated in Fig. 2, and model structure in Fig. 3. The module transforms source features into the semantic style of a target group's code space by using a corresponding learnable embedding to guide the channel modulation mechanism. Specifically, the embedding generates a set of channel-wise weights that modulate the features, effectively steering the translation. This design allows only the backbone to be loaded during inference, with translations performed dynamically by inserting different embeddings. The model backbone is implemented with stacked ConvNeXt blocks, and each embedding is a vector of shape 1×C'. During training, only the translator is trainable with encoder, codebook and backend frozen. In addition, we design a data balancing strategy to dynamically adjust the proportion of training data according to the loss changes of different targets to promote balanced spatial learning. Notably, when translation is required between only two groups, the model can degenerate into a standard one-to-one translator without relying on embeddings.

## 4 EXPERIMENTS

### 4.1 EXPERIMENTAL SETTINGS

**Dataset.** CodeAlign is evaluated on the simulated OPV2V dataset. OPV2V is a large-scale multi-modal cooperative V2V perception dataset collected in CARLA and OpenCDA, which contains

Table 1: Modality settings in the experiment. The network after LSS is its backbone.

|  | **m1** | **m2** | **m3** | **m4** | **m5** |
|---|---|---|---|---|---|
| Sensor Type | LiDAR | LiDAR | LiDAR | Camera | Camera |
| Encoder | Point-Pillar | Second | VoxelNet | LSS(ResNet-101) | LSS(EfficientNet) |

over 70 driving scenarios and each scenario contains multiple connected AVs (2 to 7) and each AV is equipped with 1 LiDAR and 4 monocular cameras.

**Modality settings.** Following HEAL's design for modality variation, five modalities are used in the experiments, as summarized in Tab. 1. The selected modalities include LiDAR and camera—two distinct sensor types with a significant domain gap—each equipped with dedicated encoders to ensure diversity in representation. For representation of modality isolation, $[\cdot\cdot]$ denotes modalities that are not isolated and are trained with collaborative data, while $\cdot + \cdot$ denotes the latter modality join the former modality for cooperation.

| Data Type | AP30 | AP50 | AP70 |
|---|---|---|---|
| Not Isolated | **89.96** | **88.52** | **80.88** |
| Isolated | 82.36 | 80.51 | 65.67 |

| Align Method | AP30 | AP50 | AP70 |
|---|---|---|---|
| Intra-group Alignment | **88.90** | **87.51** | **80.33** |
| Inter-group Alignment | 87.25 | 85.55 | 75.30 |

Table 2: Impact of data isolation on perceptual results of pyramid fusion (e2e training with alternative data of m1, m6).

Table 3: Performance of different align method for modalities that have collaborative data (m2, m6). Results show the necessity of intra-group alignment.

**Implementation details.** We employ the pyramid fusion network as the fusion net in backend. The adapter is implemented as a stack of four ResNet blocks with 3×3 convolutions. Unless otherwise specified, a codebook size of 16 is used across all experiments. All models are trained using the Adam optimizer with an initial learning rate of 0.001. Intra-group alignment is trained for 50 epochs, and intra-group alignment is trained for 30 epochs. Experiments are conducted on an NVIDIA GeForce RTX 3090 GPU. The weight of the cosine loss is set to 1. Training is conducted within the spatial range $x \in [-102.4m, +102.4m], y \in [-102.4m, +102.4m]$. See evaluation details in Appendix.

## 4.2 QUANTITATIVE RESULTS

**Impact of Modality Isolation.** Table 2 shows the perceptual performance of an end-to-end pyramid fusion network on modalities m1 and m6. When modalities are not isolated and co-occurring collaborative data is available, training with such data yields high performance. In contrast, under modality isolation, m1 and m6 have disjoint datasets, and the model is trained by alternately feeding data from each modality. This lack of collaborative supervision hinders effective feature alignment, resulting in a 15% drop in performance.

**Necessity of Intra-group Alignment.** Table 3 shows the perceptual performance on non-isolated modality pairs and illustrates the benefits of intra-group alignment. When m2 and m6 share a collaborative dataset, intra-group alignment leverages actual co-occurring data for feature fusion, achieving better performance than inter-group alignment, which relies solely on m6's local data to map its features to m2's space.

**Performance and Training Cost.** Table 4 reports CodeAlign's results on the OPV2V dataset across three key settings: the first two involve collaboration between two modality-isolated agents, and the third examines the integration of a new isolated modality into an existing group. In the first two set-

Table 4: Perception performance on OPV2V. TP denotes training parameters. At last column, we show the one-shot communication payload.

| Group settings | [m1]+[m2] | | | | [m2]+[m6] | | | | [m2m6]+[m1]+[m7] | | | | Payload |
|---|---|---|---|---|---|---|---|---|---|---|---|---|---|
| Metrics | AP30 | AP50 | AP70 | TP | AP30 | AP50 | AP70 | TP | AP30 | AP50 | AP70 | TP | |
| Single | 76.83 | 74.83 | 60.58 | 0 | 79.26 | 76.99 | 65.97 | 0 | 79.26 | 76.99 | 65.97 | 0 | 0 |
| Collab w/o Align | 78.38 | 77.76 | 67.18 | 0 | 72.38 | 71.73 | 65.54 | 0 | 89.22 | 87.17 | 77.2 | 0 | 32 MB |
| Late Fusion | 91.96 | 90.65 | 78.42 | 0 | 80.59 | 75.92 | 56.69 | 0 | 88.39 | 84.84 | 68.69 | 0 | 0.45 KB |
| HEAL | 93.00 | **92.47** | **87.69** | 1.0M | 86.79 | 84.79 | 76.01 | 1.8M | **92.49** | **91.19** | **85.06** | 17.0M | 32 MB |
| **CodeAlign** | **93.02** | 92.32 | 86.62 | **0.8M** | **88.48** | **86.53** | **76.35** | **0.8M** | 91.92 | 90.47 | 83.72 | **6.7M** | **32 KB** |

Table 5: Component ablation in intra-group training of group [m1m6].

| Codebook | Pyramid Loss | Fix Enc | Adapter | Cosine Loss | AP30 | AP50 |
|---|---|---|---|---|---|---|
| | | | | | **89.96** | **88.52** |
| ✓ | | | | | 87.51 | 86.32 |
| ✓ | ✓ | | | | 88.8 | 87.08 |
| ✓ | ✓ | ✓ | ✓ | | 89.51 | 88.17 |
| ✓ | ✓ | ✓ | ✓ | ✓ | **89.62** | **88.31** |

tings, CodeAlign achieves the best perception performance in terms of AP30, outperforming both HEAL and late fusion. Notably, in the $[m2] + [m6]$ pair, it get the highest score across APs with compressed features, even surpassing HEAL, which is Lossless end-to-end training. In the second setting, CodeAlign matches HEAL's performance with only 40% of the training cost. Moreover, while HEAL transmits uncompressed intermediate features at 32 MB per vehicle, CodeAlign reduces communication to just 32 KB, demonstrating strong suitability for bandwidth-constrained environments. Late fusion underperforms significantly, lagging CodeAlign by 14.29% in AP70 in average. The unaligned Collab w/o Align baseline shows marginal gains in the $[m1] + [m2]$ LiDAR case but suffers performance degradation in the heterogeneous $[m2] + [m6]$ pair; it exhibits slight collaboration in setting 3 because intra-group alignment between LiDAR and camera has already been performed. On average, CodeAlign surpasses other baselines while matching HEAL's accuracy with drastically reduced training cost and 1/1024th the communication overhead, highlighting its practicality and efficiency.

**Ablation of Intra-group Alignment Strategy.** Table 5 presents an ablation study on the intra-group alignment strategy. The first row corresponds to HEAL's end-to-end training, which involves no feature compression and thus serves as an upper bound on performance. When a codebook is introduced to compress the transmitted features, performance drops by 2.2% on AP50, which is acceptable and demonstrates the strong feature extraction capability of the codebook as well as the redundancy inherent in dense features. The pyramid loss helps the fusion network recover more accurate representations. To maintain system consistency, CodeAlign fixes the encoders of each modality and inserts an adapter before the codebook to facilitate alignment, resulting in a 1.09% performance gain on AP50. Finally, the cosine-based similarity loss encourages features from different modalities to become more aligned, further improving perception. At this point, CodeAlign achieves performance close to that of the lossless end-to-end model, with a 512× feature compression ratio.

**Ablation of Inter-group Alignment Strategy.** Figure 4 compares HEAL and the three one-to-many translator implementations discussed in Section 3.2.2 in terms of both perceptual performance and training cost. The evaluation focuses on mappings among the five single-modality groups in an order of m2+m1+m3+m6+m7; for simplicity, we only report the performance of mapping from modality m2 to the other four modalities. As shown in Figure 4(a), CodeAlign's code space mapping framework achieves performance on par with HEAL while requiring only 8% of its training parameters. The one-to-one approach incurs the highest training cost among the three,

Table 6: Codebook size ablation for inter-group alignment. Perception results of [m7] when surrounded by [m2]. Codebook size 16 is the best choice.

| Codebook size | | AP30 | AP50 | CV |
|---|---|---|---|---|
| [m2] | [m7] | | | |
| 4 | 4 | 75.00 | 74.50 | 16KB |
| 16 | 16 | **84.35** | **83.32** | 32KB |
| 64 | 64 | 44.29 | 43.70 | 48KB |

as it trains a separate translator for every new group pair. Figure 4(b) illustrates how the training cost scales as the number of collaborating groups increases. HEAL's cost is dominated by encoder size — when the number of groups grows from 4 to 5, its training cost surges significantly for large encoders. The one-to-one approach exhibits quadratic growth, whereas both the multi-head and channel modulation strategies scale linearly. Notably, the channel modulation variant uses only 1% of the parameters in the conditional switching module compared to the multi-head approach (from 66 to 5.2k), while achieving 0.2% higher performance, making it more practical for real-world deployment.

**Ablation on Codebook Size.** Table 6 presents the code-space translation performance between m2 and m7 using codebooks of three different sizes. With a codebook size of 4, the representation is overly compressed, failing to capture the full spectrum of features necessary for effective alignment. Conversely, a codebook size of 64 introduces excessive complexity, making the cross-codebook mapping significantly harder and degrading performance. A size of 16 strikes an optimal balance between information compression and representational capacity, yielding the best overall results.

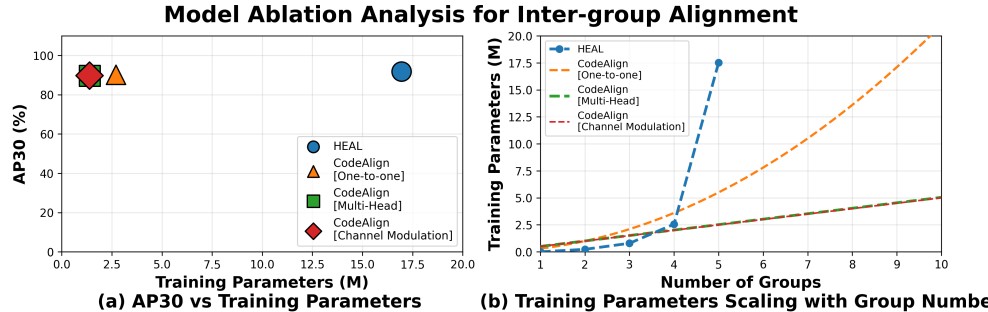

Figure 4: Ablation experiments on model structures of inter-group alignment. Test scene is [m2] mapping to 4 other single modality groups.

Comparing feature representations of different codebook sizes, shown in Figure 5 (c), (e), and (j), we observe that codebook(4) yields overly sparse and fragmented features, with poorly defined centroids. In contrast, codebook(64) introduces ambiguity and noise into the representation, diluting semantic clarity.

### 4.3 QUALITATIVE RESULTS

Figure 5 presents visualization results from the collaboratively trained group [m2,m6], illustrating codebook's effectiveness in alignment. Although camera inherently has weaker representational capacity, after alignment to the shared codebook via the adapter and subsequent decoding, it partially acquires representational characteristics, highlighted by the red boxes in subfigures (g) and (h).

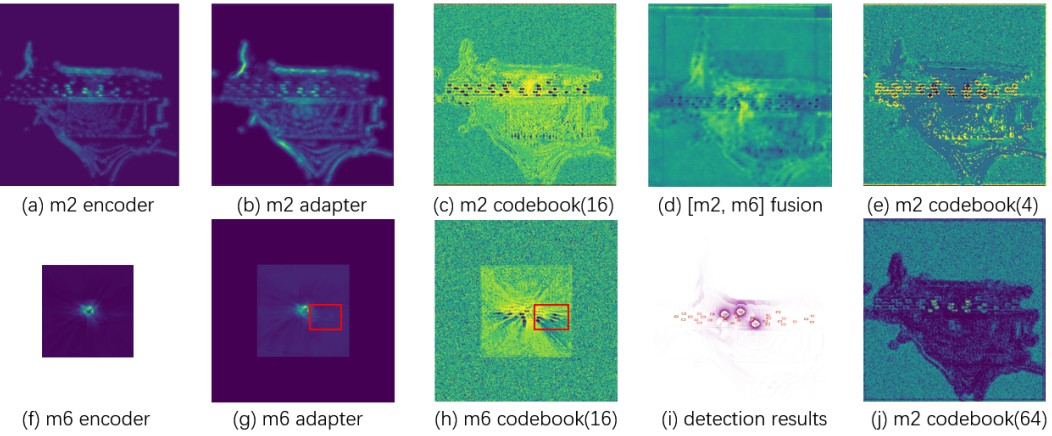

Figure 5: Visualization of BEV features during alignment, which are feature maps that have gone through the subheading module, with the size of the codebook in parentheses.

## 5 CONCLUSION

In this work, we address modality isolation — a critical yet underexplored challenge in heterogeneous collaborative perception, where the lack of co-occurring data across certain modalities exacerbates domain gaps and undermines conventional alignment strategies. To tackle this, we propose CodeAlign, the first framework that systematically resolves modality isolation through intra-group and inter-group code space alignment. By leveraging discrete code representations and a lightweight Unified Code Translator with conditional control, CodeAlign enables efficient, scalable, and communication-friendly collaboration across diverse and isolated modalities.

**Limitations.** Our evaluation is constrained by the limited modality diversity in existing simulation datasets, which prevents large-scale testing of group-wise alignment under extensive heterogeneity.

## ETHICS STATEMENT

This work is purely research-oriented and focuses on algorithmic methods for heterogeneous collaborative perception. It does not involve human subjects, personally identifiable information, or real-world deployment that could raise privacy, safety, or ethical concerns. All experiments are conducted on publicly available or simulated datasets. The proposed framework aims to improve efficiency and scalability in multi-agent perception systems and is intended for beneficial applications such as autonomous driving and smart infrastructure, with no foreseeable potential for misuse.

## REPRODUCIBILITY STATEMENT

To support reproducibility, we will release the source code, model checkpoints, and detailed implementation instructions upon publication. The code will be made publicly available on GitHub after final polishing and internal review.

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

# A APPENDIX

## A.1 EVALUATION SETTINGS

The training range is $x \in [-102.4m, +102.4m], y \in [-102.4m, +102.4m]$. In Table 4, we expand the range to $x \in [-204.8m, +204.8m], y \in [-102.4m, +102.4m]$ in evaluation for holistic view follow HEAL. We also assign the car order in OPV2V follow HEAL, that is to alternatively assign lidar and camera, to better test the heterogeneity. Note that for each test scene, for evaluation convenience, we set the first modality as the ego car.

Additionally, we design a dedicated evaluation protocol to maximally assess the alignment between two modalities. To amplify cross-modal interaction, we configure all collaborating vehicles (except the ego vehicle) to use the second modality. All other experiments are conducted under this setting, with the default communication range matching that used during training.

## A.2 EXPERIMENT OF SINGLE MODALITY GROUP FORMATION

A code space for a single modality can be constructed via several approaches, as summarized in the table. End-to-end training yields the best performance but requires modifying the original encoder architecture and incurs high training cost. Alternatively, fixing the encoder and adding an adapter achieves performance close to end-to-end training while preserving the original encoder. The most efficient and cost-effective option is to fix both the encoder and the backend and only train a lightweight adapter; although its performance is slightly lower (e.g., at the first evaluation point), it avoids fine-tuning the parameter-heavy backend, offering the best trade-off between efficiency and effectiveness.

## A.3 SINGLE MODALITY PERFORMANCE

Table 8 compares the performance of each modality with and without the codebook (size 16) in both single-agent and cooperative settings. For high-performing modalities (m1–m3, e.g., LiDAR variants), introducing the codebook leads to minimal performance change—demonstrating that the discrete code space preserves rich semantic information effectively. In contrast, for weaker modalities

Table 7: Performance of different strategies for training a codebook(16) for single group [m1]

| Strategies | AP30 | AP50 | AP70 |
|---|---|---|---|
| end-to-end retrain | **95.9** | **95.33** | **91.06** |
| fix encoder | 94.83 | 94.24 | 87.89 |
| fix encoder and add adapter | **95.22** | **94.87** | **90.47** |
| fix encoder&backend | 93.47 | 92.69 | 84.94 |
| fix encoder&backend and add adapter | 94.84 | 94.28 | 89.20 |

Table 8: Single modality performance for the 5 modalities, with or without codebook.

| | | With Codebook(16) | | | Without Codebook | | |
|---|---|---|---|---|---|---|---|
| | modality | AP30 | AP50 | AP70 | AP30 | AP50 | AP70 |
| Cooperate | m1 | 0.959 | 0.9532 | 0.915 | 0.9591 | 0.9548 | 0.9206 |
| | m2 | 0.9605 | 0.9563 | 0.9235 | 0.9596 | 0.9548 | 0.9265 |
| | m3 | 0.9581 | 0.9535 | 0.9108 | 0.9599 | 0.9554 | 0.9232 |
| | m4 | 0.5558 | 0.4773 | 0.2853 | 0.5841 | 0.5048 | 0.3149 |
| | m5 | 0.6135 | 0.5424 | 0.3606 | 0.6222 | 0.5389 | 0.3427 |
| Single | m1 | 0.8365 | 0.8171 | 0.704 | 0.8409 | 0.8258 | 0.7289 |
| | m2 | 0.85 | 0.837 | 0.7458 | 0.8347 | 0.8175 | 0.7341 |
| | m3 | 0.8344 | 0.8186 | 0.7059 | 0.8402 | 0.8242 | 0.7246 |
| | m4 | 0.2668 | 0.1841 | 0.0776 | 0.2722 | 0.1912 | 0.0808 |
| | m5 | 0.3145 | 0.2264 | 0.1126 | 0.3244 | 0.2225 | 0.1011 |

(m4–m5, e.g., camera or radar), the codebook slightly reduces absolute performance, particularly at stricter IoU thresholds (AP70), suggesting a mild compression loss. Nevertheless, the performance drop is marginal, confirming that the codebook-based representation remains faithful to the original features while enabling efficient cross-modal alignment. Overall, the results validate that code space formation incurs negligible degradation in perception quality, making it a viable foundation for heterogeneous collaboration.

# B  LLM USAGE

The authors used a large language model (LLM) solely for language polishing and grammatical refinement of the manuscript. All technical content, experimental design, analysis, and conclusions were conceived and produced independently by the authors without any assistance from the LLM.

