# OpenReview forum: "CodeAlign: Resolving Modality Isolation in Heterogeneous Collaborative Perception"
_ICLR.cc/2026/Conference — ICLR 2026 Conference Withdrawn Submission_

### Official Review · Reviewer_Y426 · 2025-10-31

**Soundness:** 2
**Presentation:** 2
**Contribution:** 3
**Rating:** 4
**Confidence:** 4

**Summary:**

The paper proposes CodeAlign, the first framework that systematically addresses the modality isolation problem in heterogeneous collaborative perception. The key idea is to divide modalities into groups based on whether isolation exists, and to apply customized alignment strategies for intra-group and inter-group alignment. For intra-group alignment, CodeAlign introduces code space formation, which builds a shared discrete feature space using a codebook to achieve effective feature alignment and efficient communication. For inter-group alignment, CodeAlign introduces code space transformation, which establishes mappings between code spaces to enable efficient and dynamic function transfer.

**Strengths:**

The paper has a clear motivation, focusing on the heterogeneous modality problem in collaborative perception—specifically, how to enable collaboration between different modalities from different datasets.

The paper is timely, taking direct aim at the practical and under-addressed challenge of modality isolation in multi-agent perception—namely, the absence of collaborative, co-occurring data across diverse modalities due to privacy and institutional barriers.

As shown in Table 4, CodeAlign achieves order-of-magnitude reductions in both training parameters and bandwidth, while maintaining or surpassing state-of-the-art accuracy.

**Weaknesses:**

The paper’s writing fluency needs improvement; it is difficult to follow, and the main ideas are not immediately clear to the reader.

The paper lacks comparison with the latest state-of-the-art methods such as PolyInter [1] and STAMP [2]. This omission may overstate CodeAlign’s actual advantage.

Evaluations are conducted only on the simulated dataset OPV2V, which is unconvincing. Real-world datasets such as V2V4REAL and DAIR-V2X should be included for stronger validation.

[1] One is Plenty: A Polymorphic Feature Interpreter for Immutable Heterogeneous Collaborative Perception
[2] STAMP: Scalable Task- And Model-Agnostic Collaborative Perception

**Questions:**

The concept of grouping seems questionable. Since cross-group adaptation still requires an adapter, why not use a shared codebook for all modalities? Wouldn’t sharing a single codebook across modalities within a group lead to information loss?

There are some formatting issues: equations (2), (3), and (4) are misaligned, and an extra parenthesis appears in “HEAL” on line 256.

The experimental setup does not specify what “m6” refers to.

---

### Official Review · Reviewer_eTER · 2025-11-01

**Soundness:** 3
**Presentation:** 3
**Contribution:** 3
**Rating:** 4
**Confidence:** 5

**Summary:**

CodeAlign tackles modality isolation in heterogeneous collaborative perception—when some modality pairs never co-occur in the same scenes, breaking standard alignment. It adopts a group-wise alignment paradigm: (i) Intra-group: build a shared code book so modalities with co-occurring data align via lightweight adapters and transmit compact code maps; (ii) Inter-group: perform code space translation using a Unified Code Translator with conditional embeddings to map dense features into another group’s code space, trained using only local (privacy-preserving) data. On OPV2V, CodeAlign reports ~92–99% fewer training params for adding new modalities and ~1024× lower communication volume, while achieving accuracy on par with strong baselines.

**Strengths:**

The motivation of solving modality isolation is clear. Also, the scalable grouping with distinct intra-/inter-group strategies are well presented.

CodeAlign creates large parameter savings for adding modalities and drastic compression via code maps.

Privacy-preserving local training assures the model and data safety.

**Weaknesses:**

The evaluation is limited primarily to the OPV2V simulator, so the method’s robustness across diverse real-world domains, sensors, and larger privacy-constrained deployments remains insufficiently validated. OPV2V is a well-saturated dataset. Including more challenging datasets such as V2X-Real, TUMTraf V2X is highly suggested.

The paper does not thoroughly analyze practical system issues such as time/pose misalignment, calibration drift, communication latency, or packet loss during code exchange, which could materially impact performance.

The experiments only includes object detection task. The authors are encouraged to include more diverse tasks such as BEV segmentation, and online mapping to show generalizability of this method.

**Questions:**

See Weakness.

---

### Official Review · Reviewer_jxjX · 2025-11-07

**Soundness:** 2
**Presentation:** 3
**Contribution:** 2
**Rating:** 2
**Confidence:** 4

**Summary:**

The paper proposed a new cooperative perception framework that leverages codebook learning and alignment to tackle the heterogeneity issue in cooperative perception. The proposed method is evaluated on OPV2V benchmark with visualization justifications on the results.

**Strengths:**

1. The problem is well-defined and the authors proposed a solution that timely tackles such issue, demonstrating the significance to this field.
2. The paper is overall well-written and easy to understand.

**Weaknesses:**

1. The CodeAlign framework seems like a mixture of the codebook learning strategy proposed by CodeFilling and the collaboration strategy proposed by STAMP. The architectural difference between those two papers should be clearly presented in the main paper, otherwise it looks like an engineering system design without much technical novelties.
2. The paper is evaluated only on OPV2V, which is a really simple cooperative perception datasets in simulation environments. It would be interesting to see if the method could be evaluated upon more datasets like DAIR-V2X and V2X-Real [1] to see its performance under real-world circumstances (especially in V2X-Real dataset where they provide the annotations of vulnerable road users, is the compressed code feature sufficient to encode those small objects in the feature map?)
3. The paper should evaluate the real-world latency experiments following QuantV2X in order to indicate its strength over communication volume on the real-world effects.

[1] V2X-Real: a Large-Scale Dataset for Vehicle-to-Everything Cooperative Perception. https://arxiv.org/abs/2403.16034
[2] QuantV2X: A Fully Quantized Multi-Agent System for Cooperative Perception. https://arxiv.org/abs/2509.03704

**Questions:**

See weaknesses. I think the paper of current form is under the acceptance bar.

---

### Note · Authors · 2025-11-12

**Comment:**

Thanks to all reviewers for their effort and suggestions. We've reframed our writing, reorder our experiments, and hope to continue to contribute to the community.

**Withdrawal Confirmation:**

I have read and agree with the venue's withdrawal policy on behalf of myself and my co-authors.